# The Consumption of Alcoholic Beverages and the Prevalence of Cardiovascular Diseases in Men and Women: A Cross-Sectional Study

**DOI:** 10.3390/nu11061318

**Published:** 2019-06-12

**Authors:** Edyta Suliga, Dorota Kozieł, Elzbieta Ciesla, Dorota Rebak, Martyna Głuszek-Osuch, Edyta Naszydłowska, Stanisław Głuszek

**Affiliations:** 1The Department of Nutrition and Dietetics, The Institute of Public Health, Faculty of Medicine and Health Sciences, Jan Kochanowski University, ul. Zeromskiego 5, 25-369 Kielce, Poland; 2The Department of Surgery and Surgical Nursing with the Scientific Research Laboratory, The Institute of Medical Sciences, Faculty of Medicine and Health Sciences, Jan Kochanowski University, ul. Zeromskiego 5, 25-369 Kielce, Poland; dorota.koziel@wp.pl (D.K.); dorota.rebak@ujk.edu.pl (D.R.); sgluszek@wp.pl (S.G.); 3The Department of Developmental Age Research, The Institute of Public Health, Faculty of Medicine and Health Sciences, Jan Kochanowski University, ul. Zeromskiego 5, 25-369 Kielce, Poland; eciesla@ujk.edu.pl; 4Laboratory of Medical Psychology and Education, The Institute of Public Health, Faculty of Medicine and Health Sciences, Jan Kochanowski University, ul. Zeromskiego 5, 25-369 Kielce, Poland; martynaosuch1@gmail.com; 5The Department of Social Prevention, Institute of Public Health, Faculty of Medicine and Health Sciences, Jan Kochanowski University, ul. Zeromskiego 5, 25-369 Kielce, Poland; edyta.naszydlowska@ujk.edu.pl

**Keywords:** alcohol consumption, cardiovascular diseases, men, women

## Abstract

Associations between alcohol consumption and the prevalence of cardiovascular diseases have been the subject of several studies for a long time; however, the presence and nature of any associations still remain unclear. The aim of the study was to analyze the associations between the consumption of alcoholic beverages and the prevalence of cardiovascular diseases in men and women. The data of 12,285 individuals aged 37–66 were used in the analysis. Multiple logistic regression models were utilized to estimate odds ratios and confidence intervals. The multivariable models included several potential confounders including age, education, marital status, body mass index (BMI), physical activity, smoking, coffee consumption, and statin use. The analyses were performed separately for men and women. In the model adjusted for confounders, the consumption from 0.1 to 10.0 g of alcohol/day was related to a lower risk of coronary disease and stroke (*p* < 0.05), and the consumption from 0.1 to 15.0 g/day was related to a lower risk of hypertension in women (*p* < 0.05). In men, in the adjusted model, there were no associations between alcohol consumption and the occurrence of hypertension or stroke. The risk of circulatory failure was significantly lower in the group in which participants drank more than 20.0 g of alcohol/day (*p* < 0.05) compared to nondrinkers. The risk of coronary disease was lower in drinkers at every level of alcohol consumption (*p* < 0.05) compared to nondrinkers. Alcohol consumption was related to a lower prevalence of cardiovascular diseases (CVD), both in men and women.

## 1. Introduction

Cardiovascular diseases (CVD) have been the main cause of deaths both in Europe and worldwide for many years [1], and the frequency of CVD deaths is higher in men than in women [2]. Modifiable CVD risk factors include dyslipidemias, abnormal glucose level, diabetes, obesity, malnutrition, low physical activity, tobacco smoking, chronic stress, and excessive alcohol consumption [3,4,5,6,7]. However, cardio-preventive factors involve healthy dietary patterns, including an adequate intake of fruit and vegetables, nuts, legumes and seeds, whole grains and seafood, physical activity, and moderate alcohol consumption. Research results on the influence of alcohol consumption on health are still ambiguous [8,9,10]. To date, the amount of alcohol which could be considered safe for health has not been definitively established [10,11,12,13,14,15].

Several reports indicate the beneficial effect of a low intake of alcoholic beverages on the reduction of CDV risk [16,17,18]. However, the definition of light, moderate, or heavy alcohol consumption differed substantially among studies. Kalinowski and Humphreys revealed that most national governments do not define low-risk drinking [19]. Results of some research indicate that the risk of CVD is the lowest with an intake of 2.5–14.9 g/day [13]. Another study showed a protective effect of alcohol consumption up to 72 g/day [20]. In some countries, there is also no consensus as to whether low-risk drinking should be defined in the same way for men and women [19]. The relationship between alcohol consumption and CVD risk in both sexes differs in some studies [12,21,22].

Despite a decrease in CVD mortality in recent years, Poland is still considered to be a high-risk country in relation to this group of diseases [6,23]. The highest level of CVD mortality in Poland concerns the Świętokrzyski region, which has a CVD mortality rate that is more than 26% higher than the average level in Poland [24]. The aim of the study was an analysis of associations between the consumption of alcoholic beverages and the prevalence of cardiovascular diseases in men and women from the Świętokrzyski region.

## 2. Subjects and Methods

### 2.1. Study Design and Sample Collection

The research material comprises the data of 13,172 participants of the Polish-Norwegian Study (PONS) project, aged 37–66. It was a cross-sectional study conducted from 2010 to 2012 in the Świętokrzyski region in Poland. Detailed information regarding the project, group selection, and research procedures were described in previously published papers [25,26]. Briefly, an invitation to participate in the study was addressed to all inhabitants of Kielecki Poviat in the region of Świętokrzyskie, aged between 45 and 64. The recruitment of participants was carried out through media outreach, press advertising, and leaflet distribution. The participation rate was 12%. In this study, information on chronic diseases, lifestyle, and the socio-demographic data of the participants was used, which was collected in face-to-face interviews using structured questionnaires. From the overall number of 13,172 participants, the data of 12,285 individuals (those that had complete sets of data) were used for further analysis (8199 women).

### 2.2. Ethical Approval

The study was approved by the Ethics Committee from the Cancer Centre and Institute of Oncology in Warsaw, No. 69/2009/1/2011 (data collection), and by the Committee on Bioethics at the Faculty of Health Sciences, Jan Kochanowski University in Kielce, Poland, No. 45/2016 (data analysis). 

### 2.3. Assessment of the Prevalence of Cardiovascular Diseases

Information on cardiovascular diseases and statin use was self-reported. Study participants answered the following questions: Have you ever been diagnosed with hypertension, stroke, coronary disease, or circulatory failure? Have you taken statins regularly for the last 30 days? Cardiovascular diseases were defined in accordance with the International Statistical Classification of Diseases and Related Health Problems, 10th revision (ICD-10): hypertension—I10–I15; coronary disease—I20–I25; circulatory failure (heart failure)—I50; stroke—I60–I64.

Moreover, information on the diagnosis of diabetes and treatment of hyperglycemia and hypertension was collected in interviews. 

### 2.4. Assessment of Biomarkers and Blood Pressure

The concentration of fasting glucose, cholesterol, and triglycerides was determined from blood samples. The concentration of total cholesterol was determined by means of the enzyme method with esterase and cholesterol oxidase, and LDL- and HDL-cholesterol with the use of the colorimetric non-precipitation method. The glucose concentration in the blood serum was determined by means of the enzyme method with hexokinase, and the concentration of triglycerides by means of the phosphogliceride oxidaseperoxidase method. Blood pressure was measured with the use of the blood pressure monitor Omron, (model M3 Intellisense, Mannheim, Germany). The test was carried out on the artery of the right upper limb, when seated, and the average of the two measurements was used in analyses. 

### 2.5. Alcohol Consumption

Information concerning alcohol intake was obtained using a standardized Food Frequency Questionnaire (FFQ) that included questions regarding participants’ typical consumption of beer, wine, and vodka during the previous 12 months. The PONS FFQ was constructed based on a previously developed and validated FFQ for the Poland branch of the Prospective Urban and Rural Epidemiological (PURE) study. FFQ has good validity and is reproducible in relation to the referential method [27]. For each participant, a daily intake of pure ethanol was calculated, taking into consideration mean alcohol content in specific alcoholic beverages and frequency of intake. All subjects were divided into 5 groups: abstainers (persons reporting no alcohol intake during the previous 12 months) and drinkers: 0.1–10.0, 10.1–20.0, 20.1–30.0, and >30.0 g of ethanol/day in men, and 0.1–5.0, 5.1–10.0, 10.1–15.0, and >15 g of ethanol/day in women. A diverse division into intake categories based on gender was applied due to different patterns in the alcohol consumption in men and women [19,28]. 

### 2.6. Socio-Demographic Variables, BMI, and Lifestyle Data

The socio-demographic variables included: sex, age, marital status (married or in a relationship; single or a widower) and education (total number of education years). The BMI (kg/m^2^) of participants was calculated on the basis of direct measurements of height and mass. The portion of coffee consisted of one cup (250 mL). Smoking behavior was assessed on the basis of the prevalence of current smokers, former smokers, and those who had never smoked. The respondents who smoked cigarettes on a daily basis during the study were classified as current smokers, and those who had not smoked for longer than 6 months were classified as former smokers. The rest of the participants comprised the group of nonsmokers. Physical activity (PA) was evaluated with the use of the International Physical Activity Questionnaire (IPAQ)—the long form. Total PA was calculated and expressed as metabolic equivalents (MET/min/week^−1^) [29]. 

### 2.7. Statistical Analysis

All categorical variables were reported as frequency and percentage (N, %) and all continuous variables were expressed as means and standard deviations (X ± SD) or medians and interquartile ranges (Me ± IQR). Cases of continuous variables whose distribution deviated from the normal (according to the results of Kolmogorov–Smirnov test) included factors such as age, body mass, BMI, coffee consumption, physical activity, and years of education. In order to compare the differences between the basic characteristics in the groups of men and women, the non-parametric Mann–Whitney U test with the continuity correction was used. A Chi-square test was used to calculate the relationship between sex and categorical variables: smoking (never, past, current), alcohol consumption (nondrinkers and four categories created according to daily alcohol intake separately for men and women) (Table 1), marital status (married or in a relationship/widow or single), hypertension (yes/no), stroke (yes/no), coronary disease (yes/no), circulatory failure (yes/no), statin use (yes/no), high total cholesterol (yes/no), high LDL cholesterol (yes/no), low HDL cholesterol (yes/no), high triglycerides (yes/no), type 2 diabetes (yes/no), high fasting glucose or hyperglycemia treatment (yes/no). The multivariate logistic regression analyses were used to estimate the odds ratios (ORs) and 95% confidence intervals (CIs) for the prevalence of each of the four CVDs. Nondrinkers were adopted as the reference group. The multivariable models included several potential confounders including age, education, physical activity, coffee consumption, and BMI, analyzed as continuous variables, and the following categorical variables: marital status (married or in a stable relationship—reference level/single or a widow/widower), smoking (never—reference level, past, and current), dyslipidemia (yes/no—reference level), regular statin use (yes/no—reference level), type 2 diabetes (yes/no—reference level), high fasting glucose or hyperglycemia treatment (yes/no—reference level). The analyses were performed separately for men and women. A *p* value < 0.05 was considered to be statistically significant. All data were analyzed using Statistical Package Statistica software (version 13.1, Warsaw, Poland).

## 3. Results

In the subject population, hypertension was diagnosed in 37.3%, stroke in 1.7%, coronary disease in 9.1%, and circulatory failure in 6.3% of the participants (Table 1). Statins were used by 16.5% of subjects. Circulatory failure was more common in women (*p* < 0.001), whereas strokes were more prevalent in men (*p* = 0.049). No differences were found in the prevalence of hypertension, coronary disease, and statin use dependent on sex. Elevated LDL cholesterol (Low Density Lipoproteins), triglycerides, type 2 diabetes, and high fasting glucose occurred in a higher proportion in men than in women (*p* < 0.001). Higher alcohol consumption was noted in men, whereas more abstainers were found among women (*p* < 0.001). Women drank more coffee (*p* < 0.001), smoked less frequently (*p* < 0.001), and had a statistically lower Body Mass Index (BMI) than men (Me ± IQR = 27.3 ± 6.4 kg/m^2^ vs. 28.2 ± 4.9 kg/m^2^). However, there were no physical activity (PA) differences based on gender. There were more married individuals, as well as those in stable relationships, among men than among women (*p* < 0.001). Women were characterized by a significantly longer period of education compared to men (Me ± IQR = 13.0 ± 5.0 years vs. 12.0 ± 5.0 years).

In the unadjusted model, at each level of alcohol consumption, the risk of hypertension and coronary disease was significantly lower in women who drank compared to abstainers (Appendix A). A smaller risk of circulatory failure was observed in women drinking between 0.1 and 15 g alcohol/day, and the risk of stroke was lower in the group of those drinking between 0.1 and 10 g alcohol/day. The higher the consumption of alcohol in men, the lower the risk of coronary disease. The lowest risk of circulatory failure was noted in the group of men drinking 10.1–30.0 g alcohol/day, and the smallest risk of hypertension among those drinking between 20.1–30.0 g alcohol/day. The risk of stroke in men was not significantly related to alcohol consumption; moreover, in the group of the highest intake, not one case of stroke was reported.

In the model adjusted for all confounders, no significant associations between alcohol consumption and circulatory failure were found in women (Table 2). In the case of coronary disease and stroke, no more significantly smaller risks were determined at the two highest levels of alcohol consumption present in the unadjusted model. The risk of hypertension was significantly lower in women drinking between 0.1 and 15.0 g of alcohol than that in abstainers. In men, in the adjusted model, no connection was found between alcohol consumption and the occurrence of hypertension and stroke, and the risk of circulatory failure was significantly lower in the group of individuals drinking more than 20.0 g of alcohol/day. However, it was noted that the risk of coronary disease was lower than in abstainers at every level of alcohol consumption, similar to the unadjusted model.

## 4. Discussion

Associations between alcohol consumption and the prevalence of CVDs have been the subject of several studies for a long time; however, they still remain unclear. In the subject population, low consumption of alcohol (from 0.1 to 10 g/day) was found to be related to a smaller risk of coronary disease and stroke, and an intake between 0.1–15 g/day was related to a lower risk of hypertension in women. Associations between alcohol consumption and circulatory failure were only present in the unadjusted model. In men, the risk of coronary disease at each level of alcohol consumption was lower compared to abstainers. The risk of circulatory failure was significantly lower in the group of drinkers consuming more than 20.0 g of alcohol/day, whereas there was no association between alcohol consumption and the occurrence of hypertension and stroke in the adjusted model. The obtained results are, to a large extent, in compliance with the findings published by other authors, indicating a lower risk of CVD in people consuming moderate amounts of alcohol compared to abstainers [13,30,31]. Bell et al. noted that in adults aged ≥30 without CVD at baseline, moderate alcohol consumption (3 units/day for men and 2 units/day for women) was connected with a lower risk of initial presentation with several diseases, such as myocardial infarction, unstable angina, heart failure, ischemic stroke, and peripheral arterial disease [30]. An analysis of the material obtained in the study of cardiovascular disease determinants within the European Prospective Investigation into Cancer (EPIC-CVD) revealed that alcohol consumption was inversely associated with non-fatal coronary heart disease risk and positively associated with the risk of different stroke subtypes [32]. In a meta-analysis of prospective studies, it was found that in the case of different CVD subtypes (apart from myocardial infarction), there were no apparent risk thresholds below which a lower alcohol consumption ceased to be related to a lower disease risk [18]. A similar phenomenon was observed in our own studies in men with coronary disease.

The cardioprotective effect of moderate alcohol consumption in relation to CVD involves a beneficial influence on reverse cholesterol transport, systemic inflammation and oxidative stress, endothelial function and platelet aggregation, reduction of body fat, improvement of insulin sensitivity, and modulation of gene expression involved in inflammation and cholesterol synthesis [33,34,35].

Differences in CVD prevalence based on alcohol consumption between men and women have not been fully explained. Polen et al. state that the unfavorable effect of alcohol on health is stronger in women than in men [36]. The abovementioned effect probably results from different alcohol metabolism rates in both sexes caused by hormonal differences. Moreover, women have a lower activity of gastric alcohol dehydrogenase and a smaller content of water in the body, which results in a higher blood ethanol concentration [37,38]. However, Ronksley et al. suggest that an increased alcohol intake was associated with a reduced risk of coronary disease in men and women, but there was no significant difference in the effect of alcohol intake and subsequent coronary disease risk between men and women [13]. In a Danish study, it was noted that coronary disease risk was lowered by 55% in women with moderate alcohol intake compared to men [39]. Zheng et al., comparing individuals with low or heavy alcohol consumption to abstainers, did not observe any differences in relative risks for cardiac death, coronary disease, or stroke between men and women [12]. They only noted that women who declared moderate alcohol consumption had a greater risk of overall mortality by 10% compared to men. Bell et al. did not find an increased heart failure risk in women who were nondrinkers, compared to those drinking moderately [30].

In the male subject population, after adjusting for confounders, significantly fewer associations between alcohol consumption and CVD risk were observed, in comparison to women. A positive effect of alcohol on CVD risk in women was present at the lowest number of units. A lower risk of hypertension was observed in women drinking from 0.1 to 15 g/day. Fisher et al. confirmed that in women with well-controlled hypertension, consuming a small amount of alcohol can be beneficial [40].

In the study population, there was no increased CVD risk in the highest categories of alcohol consumption, which was observed by several other authors [41,42]. However, this may result from the fact that a relatively low intake of alcoholic beverages prevailed among the study participants, both in men and women (Me ± IQR = 7.4 ± 12.0 g/day vs. 1.5 ± 2.8 g/day). Roerecke and Rahm noted a maximum cardioprotective influence in the case of ischemic heart disease (IHD) incidence at an average intake of 69 g alcohol/day in men and at an intake of 11 g/day in women [21]. Corrao et al. observed a cardioprotective effect of alcohol in relation to IHD at an intake of up to 72 g/day, and an increased IHD risk only at an intake of ≥89 g/day [20]. Reports concerning alcohol consumption confirm that its intake in the Świętokrzyski region did not vary from average consumption countrywide [43]. An increased mortality rate due to CVD in this region [24] seems to be an effect of other factors rather than alcohol abuse.

### Limitations

Limitations of this study involve, firstly, the cross-sectional design of our study. Thus, we cannot draw any causal inferences regarding the association of alcohol consumption with CVD. However, Knott et al. have reported the stability of alcohol consumption across the life-course among both sexes [44]. Secondly, individuals participating in surveys may sometimes have a problem with the precise determination of the amount and frequency of consumed alcoholic beverages, especially if they drink irregularly. The declared amount and the frequency of alcoholic beverage intake also could have been decreased by some respondents who abuse alcohol. Another limitation may be the fact that we were not able to separate recent abstainers from lifetime abstainers among nondrinkers. However, results of some studies revealed that a J-shaped correlation between alcohol consumption and CVD risk results is not only due to the wrong classification of abstainers comprising a reference group, i.e., including former drinkers into this group [14,21]. In addition, the study was conducted in the Swietokrzyskie region, Poland; therefore, the results obtained may not sufficiently reflect the associations that are typical of the whole country or of other populations.

The strength of this study involves the large number of participants (over 12,000), as well as for allowing for a large number of confounders in the analysis.

## 5. Conclusions

The results of the conducted study show that moderate alcohol consumption is related to lower CVD prevalence, both in men and women. However, the strength of these associations is different for specific disease entities. The lower risk of CVD in women was observed at the consumption of no more than 15 g of alcohol a day. The risk of coronary heart disease in men was lower at each alcohol intake level compared to nondrinkers, and the lower risk of circulatory failure was observed upon consumption of more than 20 g of alcohol a day. In order to prevent cardiovascular diseases in men and women more effectively, it is necessary to conduct further studies on the influence of alcohol consumption on specific disease entities.

## Figures and Tables

**Table 1 nutrients-11-01318-t001:** Sociodemographic characteristics and lifestyle habits of the subject group.

Continuous Variables		Men (N = 4086)	Women (N = 8199)	*p*
		X ± SD	X ± SD	
		Me ± IQR	Me ± IQR	
Age (years)		55.95 ± 5.42	55.54 ± 5.35	<0.001
		56.00 ± 8.00	56.00 ± 9.00	
Education (years)		13.22 ± 3.19	13.24 ± 3.17	0.026
		12.0 ± 5.0	13.0 ± 5.0	
Body mass (kg)		85.49 ± 13.04	71.50 ± 12.84	<0.001
		84.70 ± 16.50	70.00 ± 16.40	
Body Mass Index BMI (kg/m^2^)		28.47 ± 3.94	27.98 ± 4.96	<0.001
		28.17 ± 4.94	27.29 ± 6.42	
Coffee consumption (portion/day)		1.74 ± 1.90	1.89 ± 1.81	<0.001
		1.00 ± 4.36	1.00 ± 4.43	
Physical activity (Metabolic Equivalents – MET/min/week^−1^)		4633.9 ± 3952.0	4431.4 ± 3473.9	0.867
		3924.0 ± 5346.0	3217.0 ± 4607.5	
Categorical variables		N (%)	N (%)	*p*
Marital status	Married or in a relationship	3640 (89.08)	6102 (74.42)	<0.001
	Single or a widow/widower	446 (10.92)	2097 (25.58)	
Hypertension	Yes	1573 (38.50)	3013 (36.75)	0.059
	No	2513 (61.50)	5186 (63.25)	
Stroke	Yes	81 (1.98)	123 (1.50)	0.049
	No	4005 (98.02)	8076 (98.50)	
Coronary disease	Yes	396 (9.69)	719 (8.77)	0.094
	No	3690 (90.31)	7480 (91.23)	
Circulatory failure	Yes	214 (5.24)	563 (6.87)	<0.001
	No	3872 (94.76)	7636 (93.13)	
Statin use	Yes	670 (16.40)	1358 (16.56)	0.816
	No	3416 (83.60)	6841 (83.44)	
High total cholesterol (>190 mg/dL)	Yes	2602 (63.68)	5797 (70.70)	<0.001
	No	1484 (36.32)	2440 (29.30)	
High LDL cholesterol (Low Density Lipoproteins) (>115 mg/dL)	Yes	2502 (61.23)	5108 (62.30)	0.251
	No	1584 (38.77)	3091 (37.70)	
Low HDL cholesterol (High Density Lipoproteins) (<40 mg/dL in men; <46 mg/dL in women)	Yes	509 (12.46)	853 (10.40)	0.001
	No	3577 (87.54)	7346 (89.60)	
High triglycerides (>150 mg/dL)	Yes	1167 (28.56)	1531 (18.67)	<0.001
	No	2919 (71.44)	6668 (81.33)	
Type 2 diabetes	Yes	300 (7.34)	420 (5.12)	<0.001
	No	3786 (92.66)	7779 (94.88)	
High fasting glucose (≥100 mg/dL) or hyperglycemia treatment	Yes	1856 (45.42)	2243 (27.36)	<0.001
	No	2230 (54.58)	5956 (72.64)	
Smoking status	Nonsmokers	1451 (35.51)	4312 (52.59)	<0.001
	Current smokers	880 (21.54)	1504 (18.34)	
	Former smokers	1755 (42.95)	2383 (29.06)	
Alcohol consumption (g/day)				
Men	Women			
Abstainers	Abstainers	554 (13.56)	1439 (17.55)	<0.001
0.1–10.0	0.1–5.0	2235 (54.70)	5760 (70.25)	
10.1–20.0	5.1–10.0	757 (18.53)	683 (8.33)	
20.1–30.0	10.1–15.0	305 (7.46)	153 (1.87)	
>30.0	>15.0	235 (5.75)	164 (2.00)	

Note: N = number of participants; X ± SD = arithmetic mean ± standard deviation; Me ± IQR = median ± interquartile range.

**Table 2 nutrients-11-01318-t002:** Odds ratios and 95% confidence intervals for cardiovascular diseases adjusted for age, marital status, education, smoking, coffee consumption, physical activity, BMI, dyslipidemia, statin use, type 2 diabetes, and high fasting glucose or hyperglycemia treatment.

CVD	Hypertension		Stroke		Coronary Disease		Circulatory Failure	
Men								
alcohol consumption	OR	*p*	OR	*p*	OR	*p*	OR	*p*
	(95% CI)		(95% CI)		(95% CI)		(95% CI)	
abstainers (ref.)	1.00		1.00		1.00		1.00	
0.1–10.0 g	1.01	0.913	0.70	0.211	0.57	<0.001	0.74	0.113
	(0.82–1.24)		(0.39–1.23)		(0.43–0.75)		(0.51–1.07)	
10.1–20.0 g	0.90	0.387	0.55	0.118	0.50	<0.001	0.67	0.095
	(0.70–1.15)		(0.26–1.17)		(0.35–0.72)		(0.42–1.07)	
20.1–30.0 g	0.66	0.013	0.37	0.116	0.30	<0.001	0.28	0.005
	(0.48–0.92)		(0.11–1.28)		(0.17–0.54)		(0.12–0.68)	
>30.0 g	0.95	0.782	0.13	0.052	0.18	<0.001	0.36	0.014
	(0.68–1.34)		(0.02–1.02)		(0.08–0.38)		(0.16–0.81)	
Women								
abstainers (ref.)	1.00	*p*	1.00	*p*	1.00	*p*	1.00	*p*
0.1–5.0 g	0.80	0.001	0.50	0.001	0.66	<0.001	0.90	0.339
	(0.70–0.91)		(0.33–0.74)		(0.55–0.80)		(0.72–1.12)	
5.1–10.0 g	0.77	0.018	0.41	0.045	0.64	0.017	0.72	0.109
	(0.62–0.96)		(0.17–0.98)		(0.45–0.92)		(0.48–1.08)	
10.1–15.0 g	0.59	0.017	0.67	0.668	0.48	0.090	0.51	0.154
	(0.38–0.91)		(0.16–2.84)		(0.20–1.12)		0.20–1.28)	
>15.0 g	0.75	0.148	0.28	0.219	0.54	0.104	0.84	0.633
	(0.51–1.11)		(0.04–2.11)		(0.25–1.14)		(0.41–1.71)	

Note: OR = odds ratio; CI = confidence intervals; ref. = reference level; CVD = cardiovascular diseases.

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
