# Peer review of "The Consumption of Alcoholic Beverages and the Prevalence of Cardiovascular Diseases in Men and Women: A Cross-Sectional Study"

_nutrients, 2019, doi:10.3390/nu11061318_

Round 1
Reviewer 1 Report
The authors analyzed the data from 12,285 subjects and tried to understand the relationship between alcohol consumption and the prevalence of cardiovascular disease in men and women separately. Adjusted multivariate logistic regression analyses showed that a low consumption of alcohol (0.1-10g/day) reduced the risk of getting coronary disease, stroke and hypertension, but not circulatory failure in women. In men, the risk of coronary failure was reduced in all drinkers and the risk of circulatory failure only reduced in the group of drinker consuming 20.1-30.0 g alcohol/day. There is no relationship between alcohol consumption and prevalence of hypertension and stroke. It will be more clear if you could make these modifications.
Major comments:
1. It is mentioned that “the frequency of CVD deaths is higher in men than in women” in the introduction section. Is the frequency of CVD higher in women vs. in men within the group you studied?
2. It is mentioned “moderate alcohol consumption” several times in the text. However, it is not clear what is the range of moderate alcohol consumption? Is there any reference to define it?
3. It is confused if you showed the unadjusted results because it is more reliable when using adjusted results. In addition, comparing to the education and marital status, glucose levels, diabetes and lipid metabolism are more related to CVD through the introduction. Can you explain that why you did not include these factors?
Minor comments:
1. please add the citation to page 3 line 91.
2. Can you explain that the number inside the bracket for coffee consumption (1.90 for men and 1.81 for women) and physical activity (3952.0 for men and 3473.9 for women in Table 1.
3. Add “in women” to the second sentence of the first paragraph in the discussion section.
Author Response
Major comments:
1. It is mentioned that “the frequency of CVD deaths is higher in men than in women” in the introduction section. Is the frequency of CVD higher in women vs. in men within the group you studied?
The frequency of CVD in women compared to men in the study group is given in Table 1.
2. It is mentioned “moderate alcohol consumption” several times in the text. However, it is not clear what is the range of moderate alcohol consumption? Is there any reference to define it?
The definition of "moderate" alcohol consumption is unfortunately different in various countries and in studies. It is often determined based on the distribution of alcohol intake in a given population. We have added an explanation of this problem to the introduction.
3. It is confused if you showed the unadjusted results because it is more reliable when using adjusted results. In addition, comparing to the education and marital status, glucose levels, diabetes and lipid metabolism are more related to CVD through the introduction. Can you explain that why you did not include these factors?
The table containing the unadjusted results has been moved to the supplementary materials. A statistical analysis was performed again, in which glucose levels, diabetes and dyslipidemia were additionally included in the adjusted model.
Minor comments:
1. please add the citation to page 3 line 91.
done
2. Can you explain that the number inside the bracket for coffee consumption (1.90 for men and 1.81 for women) and physical activity (3952.0 for men and 3473.9 for women in Table 1.
done
3. Add “in women” to the second sentence of the first paragraph in the discussion section.
done
Reviewer 2 Report
Authors examined associations between alcohol intake and CVD prevalence (Hypertension, stroke, CHD, and circulatory failure) in a cross-sectional study. As stated in the introduction, the association may not be simple and should be studied carefully considering multiple confounders. They used a dataset including sufficient number of participants, however, there are several concerns.
Major points
1. Participation rate was low (12%) and objectives of the survey announced while recruiting. Possibility cannot be denied that people with specific characteristics (ie., using antihypertensive drug, problem drinker, etc.)
2. Definitions of outcomes (hypertension, stroke, CHD, and circulatory failure) was based on self-report. They were not ensured by BP measurement, diagnosis using CT scan (stroke) or cardiac catheter (CHD), or diagnostic criterion (circulatory failure). If a participant did not attend health check-ups, he/she will never know if he/she is hypertensive or not. Self-reported stroke, CHD, circulatory failure might be some symptoms manifested from other disease. So, the prevalence of the outcomes may be underreported or overreported.
3. Ex-drinkers (people previously had drinking habit, but stopped drinking) were not detected in the survey. It is well-known that people who had quitted drinking because of health problem have bad health (higher BP, history of diseases). The phenomenon is known as reverse causality which has to be considered in an epidemiological study.
Author Response
1. Participation rate was low (12%) and objectives of the survey announced while recruiting. Possibility cannot be denied that people with specific characteristics (ie., using antihypertensive drug, problem drinker, etc.)
Participation rate is not very high indeed. We believe that the large sample size fully compensates for the relatively low reporting rate.
It should be mentioned that the participants of our research were not recruited in a special way (personal e-mail invitation / telephones / reminder). The project assumed full voluntary participation from them. It should also be noted that the sample we tested was more than 10,000. It can be assumed that the confidence in the results obtained by us was high (error for fraction = 0.010). However, it was not too big, so we could avoid overinterpreting the data (Kaplan et al. 2014: https://www.ncbi.nlm.nih.gov/pmc/articles/PMC5439816/). In addition, there are no uniform standards regarding the necessary participation rate in scientific research.
As participants volunteered for the study, it cannot really be ruled out that people with specific health problems were more likely to attend. However, detailed data including the subjects’ pharmacological treatment of hypertension or dyslipidemia were collected in interviews, and were taken into account in the analysis.
2. Definitions of outcomes (hypertension, stroke, CHD, and circulatory failure) was based on self-report. They were not ensured by BP measurement, diagnosis using CT scan (stroke) or cardiac catheter (CHD), or diagnostic criterion (circulatory failure). If a participant did not attend health check-ups, he/she will never know if he/she is hypertensive or not. Self-reported stroke, CHD, circulatory failure might be some symptoms manifested from other disease. So, the prevalence of the outcomes may be underreported or overreported.
The occurrence of many of the analyzed diseases and health problems (e.g. hypertension, dyslipidemia, high fasting glucose) has been confirmed in medical examinations. Information on this subject has been supplemented to the work. The study was planned as screening, so it was not possible to perform computed tomography and other costly tests with over 12,000 people.
3. Ex-drinkers (people previously had drinking habit, but stopped drinking) were not detected in the survey. It is well-known that people who had quitted drinking because of health problem have bad health (higher BP, history of diseases). The phenomenon is known as reverse causality which has to be considered in an epidemiological study.
Information regarding this was included and discussed in the limitations of the study.
Reviewer 3 Report
In this study, the authors evaluated the association of alcohol intake with prevalence of cardiovascular diseases in a large population sample of Polish men and women. In men, the risk of coronary disease was lower in drinkers at every level of alcohol consumption (p<0.05) compared to nondrinkers. In women, the low-moderate consumption of alcohol (less than 15.0 g/day) was related to a lower risk of hypertension, stroke and coronary disease (p<0.05)< span="">
The main strenght is the large sample size.
However, some limitations have to be addressed:
- a linguistic revision is required
- the observational study design cannot allow to conclude that a moderate alcohol consumption has a cardioprotective effect, so no recommendations can be given on the basis of these results. Conclusions should be revised.
- introduction should better explain which is the "point of discussion" in literature, giving the readers major information about conflicting and ambigous results
- it is not clear in methods which was the age- criterion of selection (37-66 years or 45-65 years?)
- what does "circulatory failure" mean?
- information concerning alcohol intake was obtained using a standardized questionnaire. Was it validated? Please, report the reference
- table 2 is poor informative and could be eliminated
- this study is performed in a population sample of inhabitants of Kielecki Poviat in the region of Świętokrzyskie. Generalizability of results should be discussed as limitation
Author Response
- a linguistic revision is required
The article has been proof-read (MDPI English Editing Service)
- the observational study design cannot allow to conclude that a moderate alcohol consumption has a cardioprotective effect, so no recommendations can be given on the basis of these results. Conclusions should be revised.
The conclusion has been changed.
- introduction should better explain which is the "point of discussion" in literature, giving the readers major information about conflicting and ambigous results
The introduction has been completed as suggested by the reviewer.
- it is not clear in methods which was the age- criterion of selection (37-66 years or 45-65 years?)
The project originally was to study people aged 45-65, but the willingness to participate in the study was also expressed by a fairly large group of younger people, as well as a certain number of older people. Ultimately, we decided to include in the study, as well as in further analysis, people aged 37-66, so that the group of participants covered a range not greater than 30 years.
- what does "circulatory failure" mean?
Heart failure
Diseases that appear in the texts are defined in accordance with the International Statistical Classification of Diseases and Related Health Problems, 10th revision (ICD-10)
- information concerning alcohol intake was obtained using a standardized questionnaire. Was it validated? Please, report the reference
Supplemented as suggested by the reviewer.
- table 2 is poor informative and could be eliminated
Table 2 has been removed from the main text and included as additional material (supplementary file 1)
- this study is performed in a population sample of inhabitants of Kielecki Poviat in the region of Świętokrzyskie. Generalizability of results should be discussed as limitation
done
Reviewer 4 Report
Suliga et al. used the large-scale human data to investigate the association of the alcohol consumption with cardiovascular diseases in men and women. It is certainly an important study as a critical reference to similar studies in the literature, but more methodological details should be provided in this manuscript before the further review of the conclusions.
1. The clinical standards of defining the cardiovascular diseases, such as hypertension and stroke, in the study should be provided. The number of the patients with each cardiovascular disease should also be given.
2. Can the authors provide the reason and more details of using Chi-square test and the U Mann-Whitney-test to calculate the differences in different baseline characteristics? More methodological details should be included in the Statistical Analysis section. For example, what the detailed categories were defined in Chi-square test in which parameter? The Age parameter (mem: 55.95 ± 5.42 vs women: 55.54 ± 5.35) seems not quite different considering the large sample size, but the p-value is<0.0001. Some other parameters showed similar problems. I would suggest the authors to check if all statistical approaches are appropriate and ensure the correct analyses in all the parameters.
3. It is difficult to understand the calculated values and their ranges in Table 1 even though the authors mentioned in the Method: “All categorical variables were reported as frequency and percentage (N, %) and all continuous 104 variables were expressed as means and standard deviations (X±SD) or medians and interquartile 105 ranges (Me±IQR).” For example, why are the percentage values of Hypertension N (%) more than 1000? More descriptions of the methods and results will be helpful to the readers.
4. Tables 2 and 3 should be better organized and presented, e.g., using table legends. It is difficult to follow all the statistical values.
5. Representative figure presentations (e.g., alcohol consumption comparisons) of the most important findings in the tables would be better to deliver the results.
6. More details regarding the methods of unadjusted or adjusted models should be provided.
7. Try to avoid using “0.000” in p-values.
Author Response
1. The clinical standards of defining the cardiovascular diseases, such as hypertension and stroke, in the study should be provided. The number of the patients with each cardiovascular disease should also be given.
Cardiovascular diseases were defined in accordance with the International Statistical Classification of Diseases and Related Health Problems, 10th revision (ICD-10)
The number of patients with each cardiovascular disease is given in Table 1.
2. Can the authors provide the reason and more details of using Chi-square test and the U Mann-Whitney-test to calculate the differences in different baseline characteristics? More methodological details should be included in the Statistical Analysis section. For example, what the detailed categories were defined in Chi-square test in which parameter? The Age parameter (mem: 55.95 ± 5.42 vs women: 55.54 ± 5.35) seems not quite different considering the large sample size, but the p-value is<0.0001. Some other parameters showed similar problems. I would suggest the authors to check if all statistical approaches are appropriate and ensure the correct analyses in all the parameters.
Inaccuracies regarding the use of both tests are explained in the "statistical analysis" section. It has been specified for which variables the chi-square test and for which the U-Mann-Whitney test were used. All parameters and analyzes have been thoroughly checked and verified in terms of correctness of performance and were correctly made. The age distribution was assessed using the Kolmogorov-Smirnov test with the Lilieforce amendment and we found that it differed significantly from the normal distribution. Therefore, in the analysis we used the Mann-Whitney test with the continuity correction. The level of significance regarding the age difference between men and women is correct and results from the large group size.
3. It is difficult to understand the calculated values and their ranges in Table 1 even though the authors mentioned in the Method: “All categorical variables were reported as frequency and percentage (N, %) and all continuous 104 variables were expressed as means and standard deviations (X±SD) or medians and interquartile 105 ranges (Me±IQR).” For example, why are the percentage values of Hypertension N (%) more than 1000? More descriptions of the methods and results will be helpful to the readers.
The first given number "N" – refers to the number of participants, so it can be higher than 1000. Numbers in brackets indicate percentages and for hypertension are respectively 38.50% in men and 36.75% in women. As suggested by the reviewer, the table has been corrected and more descriptions have been added.
4. Tables 2 and 3 should be better organized and presented, e.g., using table legends. It is difficult to follow all the statistical values.
The tables have been corrected and added to the legend
5. Representative figure presentations (e.g., alcohol consumption comparisons) of the most important findings in the tables would be better to deliver the results.
We decided that the figures given in the tables are a more precise presentation of the results.
6. More details regarding the methods of unadjusted or adjusted models should be provided.
The details of the models have been supplemented in the methodological section.
7. Try to avoid using “0.000” in p-values.
Done
Round 2
Reviewer 1 Report
The author modified the manuscript and I am satisfied with the revised version.
Author Response
We thank the reviewers for their comments.
Reviewer 2 Report
Association between alcohol consumption and history of CHD or stroke cannot be examined in a cross-sectional study, because patients diagnosed with CHD or stroke usually receive intensive medical treatment both pharmaceutical and non-pharmaceutical which includes drinking habit. For that purpose, longitudinal design should be assigned.
Author Response
Association between alcohol consumption and history of CHD or stroke cannot be examined in a cross-sectional study, because patients diagnosed with CHD or stroke usually receive intensive medical treatment both pharmaceutical and non-pharmaceutical which includes drinking habit. For that purpose, longitudinal design should be assigned.
Response: We are aware that we can not draw any causal conclusions regarding the relationship between alcohol consumption and CVD. But in many scientific journals the results of cross-sectional studies are published. Information regarding this was included and discussed in the limitations of the study. In addition, Knott et al. have reported the stability of alcohol consumption across the life-course among both sexes [Knott, C.S.; Bell, S.; Britton, A. The stability of baseline-defined categories of alcohol consumption during the adult life-course: a 28-year prospective cohort study. Addiction 2018, 113, 34-43].
Reviewer 3 Report
The paper has now much improved.
Author Response

(The authors gave the same response as above.)

Reviewer 4 Report
All the previous comments have been addressed.
Author Response

(The authors gave the same response as above.)
